# Fall Armyworm-Induced Secondary Metabolites in Sorghum Defend Against Its Attack

**DOI:** 10.3390/insects16020218

**Published:** 2025-02-17

**Authors:** Juan-Ying Zhao, Qi Lu, Jiang Sun, Li-Yuan Sun, Ruiyan Ma, Yuanxin Wang, Jun Hu, Huiyan Wang, Yizhong Zhang, Dong Jia, Jun Yang

**Affiliations:** 1Sorghum Research Institute, Shanxi Agricultural University, Jinzhong 030600, China; zjy0502@yeah.net (J.-Y.Z.); why881206@163.com (H.W.); zhyzh225@163.com (Y.Z.); 2Hou-Ji Laboratory in Shanxi Province, Shanxi Agricultural University, Taiyuan 030031, China; 3College of Plant Protection, Shanxi Agricultural University, Jinzhong 030801, China; luqichuqing@163.com (Q.L.); sunjiang198@163.com (J.S.); sliyuan1314@163.com (L.-Y.S.); mary@sxau.edu.cn (R.M.); wangyuanx1992@163.com (Y.W.); hujun@sxau.edu.cn (J.H.); biodong@foxmail.com (D.J.); 4Shanxi Key Laboratory of Integrated Pest Management in Agriculture, College of Plant Protection, Shanxi Agricultural University, Taiyuan 030031, China

**Keywords:** sorghum, *Spodoptera frugiperda*, secondary metabolites, defense mechanism, antifeedant

## Abstract

In response to herbivorous insect attacks, plants have developed the ability to rapidly modify the secondary metabolisms involved in their defense against insects. The spectrum specificity of plant secondary metabolisms plays a significant role in insect adaptation to each host. The fall armyworm (FAW) *Spodoptera frugiperda* is a serious agricultural pest that has invaded China. The FAW is a polyphagous insect that feeds on gramineous crops, such as maize, sorghum, and rice. However, the chemical basis of sorghum defenses against FAWs is not well understood in the current research. This study found that sorghum can reduce FAWs’ fitness, resulting in significantly lower selection and damage rates for sorghum, and reducing larval weight relative to maize. Responding to FAW attacks, both maize and sorghum rapidly alter their secondary metabolic profiles, which show species-specific changes. Gambogenic acid and chimonanthine, which are present and induced in sorghum, have a detrimental effect on larvae feeding and growth, deterring their feeding and lowering their weight increase. These findings indicate that the present and induced secondary compounds in sorghum have a role in chemical defense against FAWs and aid in the development of new pest control strategies.

## 1. Introduction

The co-evolution between herbivorous insects and host plants is considered an arms race [1]. To prevent insects from feeding, host plants develop and form multiple defense mechanisms, such as physical and chemical ones. In response to insect feeding, plants can quickly produce a large number of chemical defense components, such as chemical signals, biotoxins (secondary metabolites), defense proteins, and so forth. Among these, plant secondary metabolites, which are poisonous defensive compounds, are the main defenses against herbivorous insects [2,3].

Plant secondary metabolites mainly consist of terpenes, phenolic compounds (phenolic acids and flavonoids), and secondary nitrogen-containing compounds (alkaloids), which play an important role in plant defense against the feeding, growth and development, and reproduction of herbivorous insects [4,5,6]. In this plant–herbivore interaction, plant defenses against insects include indirect defense and direct defense. Based on their effects on insects, these compounds function as attractants, deterrents, digestive inhibitors, and toxins [7,8]. As a type of indirect defense, attractants are attractive plant volatiles that are appealing to natural enemies. The parasitic wasps *Cotesia plutellae* of the diamondback moth can be attracted to allyl isothiocyanate (AITC), and *Dasineura brassicae* parasitoids *Omphale clypealis* and *Platygaster subuliformis* are strongly attracted to AITC and phenylethyl isothiocyanate [9]. One plant direct defense is that secondary metabolites act as deterrents, digestive inhibitors, and toxins, which directly or negatively affect the feeding, growth and development, and reproduction of insects [10]. The effect of deterrents and toxins has been studied in depth in insects. Cucurbitacin acts as a feeding deterrent for *Pieris rapae* and strongly inhibits larval feeding [11]. Similarly, judaicin 7-*O*-glucoside, 2-methoxyjudaicin, judaicin, and maackiain are four isoflavonoids that have shown antifeedant activities, deterring larval feeding by *H. armigera* [12], and saponins act as feeding deterrents for the larvae of the diamondback moth [13,14]. Previous studies have proved that certain secondary compounds are poisonous to herbivorous insects, suppressing larval growth and development and reducing the female reproduction rate. Gossypol is a crucial component of pest defense in cotton due to its toxicity [15,16,17]. Naringenin and quercetin are toxic to *Acyrthosiphon pisum*, which leads to prolonged development times, increased mortality, and decreased fecundity after feeding [18]. Resveratrol and *p*-coumaric acid are two phenolic compounds that reduce the larval weight of *Spodoptera litura* and *Amsacta albistriga* [19]. Conversely, insect feeding can boost the amount of secondary defensive compounds in plants, enhancing their ability to defend against insects. Gallic acid, 4-cinnamic acid, *p*-coumaric acid, and salicylic acid accumulate in cotton plants as a result of feeding by *H. armigera* and *S. litura*, suppressing larval weight and raising mortality [20]. *Ostrinia furnacalis* feeding increases the content of benzoxazines in maize, which decreases the relative larval growth rate and increases the relative larval consumption rate [21]. Brown planthopper (BPH), *Nilaparvata lugens*, feeding has been shown to significantly cause the accumulation of sakuranetin in the leaf sheath and phloem of rice, which has shown a detrimental effect on BPH as a sucking deterrent [22]. Therefore, discovering new bioactive defense compounds is the key to understanding the underlying defensive adaptations of host plants against insects, and the control of pests using plant secondary compounds.

The fall armyworm (FAW), *Spodoptera frugiperda*, is a notorious invasive pest in China [23,24]. The FAW is a polyphagous insect with a wide range of hosts and can feed on 353 plant species from 76 families [25]. The FAW is classed into two strains based on host preference: the maize strain and the rice strain [26,27]. The FAW found in China belongs to the maize strain [28]. In addition to maize, rice, and wheat, other gramineous grain crops, such as sorghum and millet, are the FAW’s host plants. Previous studies have demonstrated that FAW can transfer between hosts when there is a high population density or insufficient maize supplies [29]. To date, chemical control is still the main measure used to manage FAW; however, the frequent application of insecticides in a short period has led to the rapid development of insecticide resistance in this species [30]. Nonetheless, utilizing host plant resistance to defend against insects is an effective pest control strategy [31,32]. Hence, the study of the adaptation of FAW to host plants is very important for developing a new green and safe strategy to control FAWs. Plant secondary metabolites have been demonstrated to influence the feeding preferences and host adaptability of herbivorous insects [10,33]. In sorghum, some secondary compounds can reduce FAW fitness. For example, tannin and 3-deoxyanthocyanidin flavonoids in sorghum induce resistance to FAW [34,35], and the latter also contributes to resistance to *Rhopalosiphum maidis* [36]. Furthermore, it has been reported that some secondary compounds, such as cedrelone [37], carvacrol [38], essential oils from *Hyptis marrubioides* and *Ocimum basilicum* [39], and polyphenol extract present in purple maize pericarp extract [40], are toxic and/or inhibit its growth. The production of flavonoids in sorghum SC1345 is induced by FAW larvae [41]. Yet, how these compounds contribute to resistance against FAW is still unknown. Thus, plant secondary metabolites are essential for the host’s defense against herbivorous insects. However, further studies are needed to discover which secondary metabolites are present in sorghum and which secondary compounds, such as flavonoids, are involved in host defensive resistance to FAW.

Here, we investigate the underlying defense mechanism of sorghum against the invasive insect *S. frugiperda* using biological assays and non-target metabolomics. First, the preference of FAW larvae on maize and sorghum was studied by using a series of biological assays. Subsequently, non-target metabolomics was used to uncover the changes in the secondary metabolites in maize and sorghum before and after larvae infestation. Finally, the effects of sorghum-specific DSMs gambogenic acid, baohuoside II, and chimonanthine on the feeding preferences and growth and development of FAW larvae were analyzed. Our findings provide new insights into employing bioactive plant compounds against polyphagous insects and lay the foundation for the development of new strategies for pest control.

## 2. Materials and Methods

### 2.1. Insect and Plant Culture

The original fall armyworm *Spodoptera frugiperda* was collected from Dehong Dai and Jingpo Autonomous Prefecture, Yunnan Province, China. An artificial diet was prepared with 280 g maize powder, 90 g soybean powder, 35 g yeast powder, 25 g agar, 0.2 g vitamin B, 12 g cholesterol, 2 g sorbic acid, 12 g L-ascorbic acid, 5 g nipagin ester, 4 mL methanol, 0.2 g penicillin, and 1500 mL water according to the previous study [42]. The rearing conditions were a photoperiod of 16 h light–8 h darkness, a temperature of 25 ± 1 °C, and 60 ± 5% relative humidity. The pupae were separated by sex and then placed in a cage for emergence. Adults were fed with 10% honey water.

The seeds of the maize variety Zhongnongtian488 and sorghum variety Hongyingzi were planted into plastic pots (9 cm diameter × 10 cm height), respectively. All the pots were placed in a netting room under natural conditions. After germination and growth, 10 healthy and the same-sized seedlings were kept and allowed to grow for 3 weeks under natural conditions.

### 2.2. Chemicals

Gambogenic acid (≥98% purity) and baohuoside II (≥98% purity) were purchased from Yuanyeshengwu (Shanghai, China), and chimonanthine (≥98% purity) from Macklin (Shanghai, China).

### 2.3. Biological Assay

First, we used a two-choice test to evaluate the feeding preferences of the 5th-instar FAW larvae for maize and sorghum. Disks of maize and sorghum with 15 mm diameters were prepared. The maize leaf disks were used as control, and the sorghum leaf disks as the treatment. The two maize and sorghum leaf disks were alternately placed in a 9 cm diameter Petri dish. A wetting filter paper was used to cover each Petri dish to maintain humidity. Each healthy and selected larva was first starved for 2 h and then placed into the above Petri dish to feed for 2 h. The feeding area of larvae on the treated and control leaf disks was counted, and also the feeding preference index (PI) was calculated. PI = (the consumed area of the treated disk—the consumed area of the control disk)/(the consumed area of the treated disk + the consumed area of the control disk).

Second, a cage experiment was conducted to investigate the effects of different host plants on the selection preference of FAW larvae. Potted maize and sorghum seedlings were alternately placed at the four corners of a closed cage (length × width × height: 50 cm × 50 cm × 50 cm). Forty uniform 5th-instar larvae were placed in a central position at the bottom of the cage. The distance between the insect and the four corners of the cage was about 70 cm. The larvae were allowed to select host plants and to feed overnight (about 16 h) in natural conditions. The experiment was stopped after larvae feeding for 16 h, the number of maize and sorghum plants damaged by larvae and the number of larvae on the maize and sorghum plants were counted, respectively, calculating the plant damage rate and the number of larvae on the different plants. The plant damage rate (%) = (number of maize or sorghum plants damaged by larvae per pot)/(total number of maize or sorghum plants per pot) × 100.

Finally, we evaluated the effect of maize and sorghum on the growth and development of FAW larvae using a no-choice feeding assay. The newly hatched larvae were fed on fresh maize and sorghum leaves in plastic culture tubes (2.4 cm diameter × 9.5 cm height) for eight days. Larvae fed on maize leaves were used as the control group, and larvae fed on sorghum leaves as the treatment group. Each leaf was changed every day and each larva was weighed every two days. The changes in larval weight were used to evaluate the growth rate of the larvae. The control and treatment experiments were repeated five times, and five larvae were used in each replicate.

### 2.4. Sample Collection and Metabolite Extraction

Here, the potted maize and sorghum were divided into two groups: one group of maize and sorghum seedlings served as the treatment group that was infested with larvae, and the other group was considered as the control that was un-infested with larvae. For the treatment group, each potted maize or sorghum seedling was infested by about 10–15 3rd-instar larvae. After infestation, each pot of seedlings was covered with a clean plastic cylinder (9 cm diameter × 20 cm length), and its top was sealed with gauze to prevent insect escape. The soil surface of the pot was also covered with plastic wrap to prevent larvae from getting into the soil. For the control group, each potted maize or sorghum seedling was covered with a clean plastic cylinder and sealed with gauze. After 24 h, the leaves of maize and sorghum belonging to the treatment and control groups were quickly cut out, wrapped in tin foil, and then frozen in liquid nitrogen. Each control and treatment experiment on the maize and sorghum was repeated six times.

The detections of non-target metabolomics for all samples were performed by Novogene Co., Ltd., Beijing, China. All chemicals and solvents used in this experiment were obtained from Thermo Fisher Scientific (Waltham, MA, USA). Approximately 100 mg samples were individually ground with liquid nitrogen. The homogenates were resuspended with 500 μL 80% methanol–water solution by well vortexing, incubated on ice for 5 min, and then centrifuged at 4 °C and 15,000 rpm for 20 min. The supernatant was diluted with water to a final concentration containing 53% methanol and was centrifuged at 15,000 rpm for 20 min at 4 °C. The obtained supernatants were collected and filtered using a 0.22 μm filter membrane and the extracts were stored at −80 °C.

### 2.5. UHPLC-MS/MS Analysis

Ultra-high-performance liquid chromatography coupled to tandem mass spectrometry (UHPLC-MS/MS) analyses were performed using a Vanquish UHPLC system (Thermo Fisher Scientific, Waltham, MA, USA) with a Hypesil Gold column (100 × 2.1 mm, 1.9 μm) coupled to Q Exactive HF/Q Exactive HF-X Q Exactive (Thermo Fisher Scientific, Waltham, MA, USA). The binary mobile phases for the positive polarity mode were 0.1% formic acid (A) and methanol (B). The binary mobile phases for the negative polarity mode were 5 mM ammonium acetate (pH 9.0, A) and methanol (B). The solvent gradient was set as follows: 2% B, 1.5 min; 2–85% B, 3 min; 85–100% B, 10 min; 100–2% B, 10.1 min; 2% B, 12 min. The injection volume of the samples and standards was 5 µL. The column temperature was 40 °C, and the flow rate was set at 0.2 mL/min. A Q ExactiveTM HF mass spectrometer was operated in positive/negative polarity mode with a spray voltage of 3.5 kV, capillary temperature of 320 °C, sheath gas flow rate of 35 psi, and aux gas flow rate of 10 L/min, as well as an S-lens RF level of 60 and aux gas heater temperature of 350 °C.

The raw data files obtained from UHPLC-MS/MS were processed using the software Compound Discoverer 3.1 (CD3.1, Thermo Fisher Scientific, Waltham, MA, USA) to integrate and correct the peaks. Then, the peaks were matched with the mzCloud (https://www.mzcloud.org/, accessed on 5 November 2023) and ChemSpider (http://www.chemspider.com/, accessed on 5 November 2023) databases to obtain the accurate qualitative and relative quantitative result for the corresponding metabolite. The metabolites were annotated using the KEGG database (https://www.genome.jp/kegg/, accessed on 5 November 2023), HMDB database (https://hmdb.ca/, accessed on 5 November 2023), and Lipidmaps database (http://www.lipidmaps.org/, accessed on 5 November 2023). The identification of DSMs was performed by using principal components analysis (PCA), partial least squares discriminant analysis (PLS-DA), and univariate analysis (*t*-test). The metabolites with VIP (variable importance for the projection) > 1.0, FC (fold change) > 1.2, or FC < 0.833, *p*-value < 0.05, were considered to be DSMs. The heat maps of the abundance of DSMs were produced by using the heatmap tool in Hiplot Pro (https://hiplot.com.cn/, accessed on 28 October 2024).

### 2.6. Functional Assay of Candidate Compounds on S. frugiperda Larvae

The two-choice assay was carried out based on the previously reported feeding choice assay with a minor modification [43]. Gambogenic acid and baohuoside II were dissolved in acetone, and chimonanthine was dissolved in methyl alcohol. These compounds were prepared into a series of concentration gradient solutions at 20 μg/mL, 200 μg/mL, and 2000 μg/mL. Four maize leaf disks with an area of 1 cm^2^ were alternately placed in a 9 cm diameter Petri dish. The treated disk was coated with 20 μL of the candidate compounds. The concentration gradients for each compound were the same as above. The same volume of solvent (acetone or methanol) was applied as a control. Each Petri dish was covered with a humid filter paper to maintain humidity. Healthy fifth-instar larvae were first starved for 2 h and then larvae were placed in individual Petri dishes and allowed to feed on a leaf disk for 2 h. The area consumed by larvae on the treated and control leaf disks was counted for the calculation of the feeding preference index.

The effect of these three compounds on larval growth and development was evaluated by the following experiments. The artificial diet served as the substrate. Gambogenic acid, baohuoside II, and chimonanthine were prepared in 1000 μg/mL solutions. Fifty-microliter compounds were added to a 1 cm^3^ artificial diet and the same volume of acetone or methanol was added as a control. The control and treated artificial diets were put into a 24-well cell culture plate and after weighing, 2nd-instar larvae were placed in individual wells. At 3 and 5 days post-feeding, the larvae were weighed to evaluate the effect of these compounds on larval weight.

### 2.7. Statistical Analysis

Statistical analyses were performed using SPSS (SPSS 20 software, Chicago, IL, USA), and GraphPad Prism 8.3.0 (GraphPad Software, San Diego, CA, USA). Data visualization was performed by using the GraphPad Prism software. The data from the two-choice feeding experiments were analyzed with the two-tailed Student’s *t*-test. The feeding preference indexes of compound-fed larvae were analyzed by one-way ANOVA and compared with the Tukey HSD test. The analysis of the change in larval weight was applied with the nonparametric test (Mann–Whitney U test). Data are presented as the mean ± standard error of the mean (SEM). Asterisks indicate statistical significance (* *p* < 0.05, ** *p* < 0.01, *** *p* < 0.001) and n.s. shows no significant differences (*p* > 0.05). Different letters indicate significant differences according to one-way ANOVA followed by the Tukey HSD test.

## 3. Results

### 3.1. Larvae of S. frugiperda Prefer to Feed on Maize over Sorghum

Using a two-choice test, we found that FAW larvae preferred maize over sorghum (Figure 1A). The FAW larvae displayed a stronger response to maize with a feeding preference index (PI) value of 0.4 (Figure 1B). We tested the preferences of the larvae towards the maize and sorghum plants (Figure 1C). Almost all maize plants were fed on by the FAW larvae, while the sorghum plants were fed on sparingly (Figure 1D). In contrast to maize, which had a 76.7% damage rate, the sorghum plants only had 28.6% damage (Figure 1E). The number of larvae collected from the sorghum plants was also lower than that collected from the maize (Figure 1F).

Next, we studied the effects of feeding on different host plants on the growth and development of FAW larvae. The no-choice feeding assay found that feeding on sorghum reduced larval weight (Figure 2). Compared to larvae reared on maize leaves, those raised on sorghum leaves gained less mass and grew more slowly over time (Figure 2), the weight of the sorghum-fed larvae decreased by 50.3%, 62.3%, and 51.5% at 4, 6, and 8 days, respectively, in comparison to the control (maize-fed larvae) (Figure 2).

### 3.2. Screening of Differential Secondary Metabolites with Significant Changes in Maize and Sorghum

To explore the responses of host plants to the attack by insects, we detected changes in secondary metabolites in maize and sorghum both before (control) and after (treatment) FAW larvae infestation by using LC-MS/MS techniques (Figure 3A). The results showed that the response spectra of the plant secondary metabolites were altered by larvae fed on maize and sorghum. We analyzed the changes in maize secondary compounds before feeding by larvae (BZm) and after feeding by larvae (AZm) and found 25 DSMs in the BZm and AZm group, including 13 increased and 12 decreased ones (Figure 3B,C). And we discovered 57 DSMs with 34 increased and 23 decreased ones in sorghum before feeding by larvae (BSb) and after feeding by larvae (ASb) (Figure 3B,C). For these DSMs, the number of DSMs from the BZm vs. AZm group and BSb vs. ASb group were 19 and 51, respectively, and only 6 compounds were shared between them (Figure 3B and Table 1).

Phenols mainly include simple phenols and flavonoid compounds, and alkaloids are representative of secondary nitrogen-containing compounds [5]. Therefore, we went on to describe these differential compounds as terpenes, simple phenols, flavonoids, and alkaloids. In the BZm vs. AZm group, all DSMs included nine terpenes, eight simple phenols, four flavonoids, and four alkaloids, and the DSMs present in the BSb vs. ASb group were composed of twenty-eight terpenes, ten simple phenols, fourteen flavonoids, and five alkaloids (Table 1). We then further subdivided these compounds according to the classification characteristics of each class. The detailed classification of these compounds is shown in Appendix A.

### 3.3. Change in the Abundance of Differential Secondary Metabolites in Maize and Sorghum Groups

We constructed a heatmap to show the significantly induced or reduced DSMs in both the maize and sorghum groups. In the BZm vs. AZm group, six terpenes were induced in response to FAW larvae attack, with the highest degree of induction being for danshenol C (12.58 folds), followed by kaji-ichigoside F1 (4.93 folds) and cucurbitacin I (Figure 4A, Table 1). Three terpenes (pseudolaric acid B, dehydroeffusol, and rehmannioside D) showed a similar reduction pattern (Figure 4A, Table 1). However, in the BSb vs. ASb group, up to 24 terpene abundances were significantly increased, and notoginsenoside R1 showed the highest induction (12.89 folds), followed by cucurbitacin E (5.35 folds) and sclareolide (4.77 folds) (Figure 4B, Table 1) and only four terpenoids were inhibited (Figure 4B, Table 1). For the common DSMs, ingenol was significantly elevated in both groups; however, the level of rehmannioside D was reduced in the BZm vs. AZm group, but induced in the BSb vs. ASb group (Figure 4).

Larvae feeding on maize almost equally increased the abundance of five simple phenols (erianin, 2,5-dimethylphenol, protocatechuic acid, curcumin, olivetol) and decreased three (Figure 4A, Table 1). In terms of flavonoids, the only one induced was corylin. It increased by 29.11 times, and the remaining three flavonoids were inhibited by larvae feeding on the maize (Figure 4A, Table 1). In the BSb vs. ASb group, four simple phenols were increased and eight were decreased, where the abundance of martynoside was elevated by 6.44 folds, followed by 6-paradol (5.22 folds), and lupulin A (4.94 folds), and the highest degree of inhibition of a simple phenol was 7-hydroxy-4-methyl-8-nitrocoumarin (Figure 4B, Table 1). Three differential flavonoids were significantly induced and twelve were inhibited, where the abundance of gambogenic acid was increased by 6.73 folds, followed by baohuoside II (3.55 folds), and plantagoside (0.09 fold) had the most significant inhibition (Figure 4B, Table 1). In addition, the common DSMs (7-hydroxy-4-methyl-8-nitrocoumarin and 2,4-dihydroxybenzoic acid) were all reduced in both maize and sorghum groups (Figure 4).

In alkaloids, only the abundance of hordatine B was significantly increased and the remaining three alkaloids decreased in the larvae-fed maize (Figure 4A, Table 1). Conversely, in the BSb vs. ASb group, three alkaloids were significantly increased and two were inhibited, of which hordatine B (4.08 folds) displayed the highest induction degree, followed by chimonanthine (3.57 folds), and the compound reduced to the highest degree was theophylline (0.36 fold) (Figure 4B, Table 1). The common alkaloid hordatine B was induced, and theophylline was inhibited in maize and sorghum fed on by larvae (Figure 4).

Flavonoids and alkaloids have a wide range of biological functions in plants, and are important barriers for plants to defend against herbivorous insects [35,44,45]. When the larvae ingested maize and sorghum, feeding maize only caused an increase in hordatine B, but resulted in the induction of seven compounds, irisflorentin, gambogenic acid, baohuoside II, isoquinoline, chimonanthine, and hordatine B, in sorghum (Figure 4, Table 1). Therefore, we hypothesize that the DSMs are specific to sorghum and are brought on by FAW larvae feeding, particularly those that exhibit drastic alterations (FC > 3), including gambogenic acid, baohuoside II, and chimonanthine, which are involved in the defense of sorghum against *S. frugiperda*.

### 3.4. The Inhibiting Effect of Sorghum Defensive Compounds on the Larval Feeding and Growth of S. frugiperda

The larval feeding behavior was significantly inhibited by the presence of gambogenic acid and chimonanthine when gambogenic acid, baohuoside II, and chimonanthine were added to maize leaf disks (Figure 5). Additionally, the larval feeding preference index decreased as the concentration of these two compounds increased (Figure 5). However, baohuoside II had no inhibiting effect on the larval feeding, and the larval PI did not correlate with the concentration of baohuoside II (Figure 5). The inhibiting concentration of gambogenic acid and chimonanthine in FAW larvae started from 4 μg/cm^2^ (Figure 5). Compared with the larval PIs at the concentration of 40 μg/cm^2^, gambogenic acid was the most effective at inhibiting larval feeding, followed by chimonanthine (Figure 5).

We also set out to determine the effect of gambogenic acid, baohuoside II, and chimonanthine on larval growth and development. Supplemental gambogenic acid and chimonanthine in the artificial diet severely impacted growth and development (Figure 6). Baohuoside II did not decrease FAW larva body weight (Figure 6). As the feeding time was prolonged, the inhibitory effects of gambogenic acid and chimonanthine on larval weight increased gradually, despite the body weight of the larvae fed on the chimonanthine-contained diet being lower than that of the larvae fed on the gambogenic acid and baohuoside II diet (Figure 6). After three days, the weight of larvae that ingested these compounds began to decrease, but the difference was not statistically significant (Figure 6). The larval weight was significantly lower than that of the control when they were reared on a diet containing gambogenic acid and chimonanthine for 5 days, respectively (Figure 6). The larvae fed gambogenic acid and chimonanthine experienced a reduction in body weight of 34.6% and 46.2%, respectively, in comparison to the controls. In addition, we also analyzed the net growth rate of larvae from day three to day five, and the net growth rate of larvae fed with gambogenic acid and chimonanthine was significantly lower than that of the control (Appendix A).

## 4. Discussion

For herbivorous insects, seeking suitable hosts is essential for the survival and reproduction of the population. Plant chemical defense plays an important role in plant host adaptation to herbivorous insects. The defensive mechanism of sorghum, an important graminaceous crop, to the polyphagous invasive insect the FAW remains enigmatic. Elucidating how sorghum defends itself against FAWs is of great significance for the effective prevention and control of FAW damage in sorghum planting. In this study, we found that FAW preferred maize, and feeding on sorghum resulted in a decrease in larval weight. FAW larval feeding caused the change in the response spectra of the secondary metabolites in maize and sorghum, according to non-target metabolomics, and these DSMs showed species-specific distribution traits. Bioassays demonstrated that gambogenic acid and chimonanthine, which are compounds specific to sorghum, had significant inhibitory effects on larval feeding and growth.

The host range and adaptability of insects are the result of coevolution between insects and host plants. The host preferences of different insect species have evolved from adaptations to host secondary metabolites [33]. For instance, *H. armigera* is a polyphagous insect that mostly feeds on cotton and tomato; however, *H. armigera* does not prefer to feed on *Arabidopsis* because the glucosinolate in *Arabidopsis* effectively prevents larval feeding [46,47,48]. *P. rapae*, a specialist crucifer insect, prefers cabbage and broccoli over *Erysimum cheiranthoides* for feeding and oviposition because *Erysimum* contains deterrents such as cardenolide, erysimoside, and erychroside [49,50,51]. In this study, we found that sorghum can reduce FAWs’ fitness, consistent with previous reports [52,53], suggesting sorghum may contain defensive compounds that can deter FAW larvae feeding or have a toxic effect on this insect. Thus, how sorghum evolves to use secondary metabolites to defend against these herbivorous insects is worthy of further study.

Chemical defenses represented by plant secondary compounds are the key barrier of plants against insects [10,54]. In response to insect attacks, the majority of plant chemical defenses are inducible [55]. The induction of these compounds has an adverse effect on insect feeding or survival. For example, the accumulation of stigmasterol and sakuranetin in rice induced by BPH [22,56], and the production of gallic acid, 4-cinnamic acid, *p*-coumaric acid, and salicylic acid in cotton induced by *H. armigera* and *S. litura* suppress larval feeding and growth [20], Therefore, utilizing plant secondary compounds to regulate insect behavior and development is one of the most important measures for developing green pest control strategies [8,46,55]. In this study, feeding by FAW larvae changed the abundance of secondary metabolites in maize and sorghum, and these DSMs displayed species-specific changes, which indicates that in plants, altering the abundance of their metabolites is a way to respond to insect attacks. We also found that the number of DSMs in maize is lower than in sorghum. This may be a factor contributing to the preference of FAW for feeding on maize. Despite this, there is currently no evidence for the effects of induced DSMs such as danshenol C, notoginsenoside R1, martynoside, corylin, gambogenic acid, baohuoside II and chimonanthine, on insects, some studies have found that these compounds are involved in regulating some diseases and health in humans and mice [57,58,59,60,61,62,63].

Numerous induced plant secondary metabolites are directly involved in plant defense against insects. Flavonoids and alkaloids are important barriers for plants to defend against insects [35,44,45]. The contents of flavonoids and alkaloids are negatively correlated with the growth and development and survival of insects, such as sakuranetin, schaftoside, and gramine against BPH [22,44,64,65], two alkaloids, α-chaconine and α-solanine, against *Tecia solanivora* [66], cedrelone and carvacrol against FAW [37,38], and quercetin against *Cydia pomonella* [67]. We found that gambogenic acid and chimonanthine have antifeedant activity for FAW larvae, but baohuoside II has no effect. Our findings are consistent with previous reports [12,68,69]. Four isoflavonoids (judaicin 7-*O*-glucoside, 2-methoxyjudaicin, judaicin, and maackiain) displayed significant antifeedant activity for *H. armigera*. Judaicin deters *S. littoralis*, and maackiain alone can deter *S. frugiperda*. These four compounds have no deterrent effect on *Heliothis virescens* and *S. exigua* [12]. Similarly, five pyrrolizidine alkaloids (senkirkine, senecionine, seneciphylline, monocrotaline, and retrorsine) significantly decreased the survival rate of *Frankliniella occidentalis*, but heliotrine did not. None of these alkaloids inhibited *S. exigua* or *Mamestra brassicae* larvae [69]. These results imply that plant secondary metabolites have different effects for each herbivorous insect, and these metabolites play an important role in the adaptation of insects to host plants [68]. The ingestion of secondary compounds adversely affects larval growth and development, as evidenced by reduced weight. We found that feeding on gambogenic acid and chimonanthine suppressed larval weight. These observations are similar to previous reports, such as sinigrin against *H. armigera*, and pinocembrin and camptothecin against FAW [43,70,71]. The weight of larvae fed with chimonanthine was lower than that of larvae fed with gambogenic acid and baohuoside II. This may be due to solvent differences because the weight of larvae that ingested methyl alcohol was also lower than that of acetone-fed larvae. Certainly, the decrease in weight can generally be attributed to the deterrent effect of secondary compounds and/or toxicity to larvae [35,72]. Some phenolic compounds in sorghum have been implicated in the resistance to FAWs and *R. maidis* [36,41]. Therefore, these results suggest that gambogenic acid and chimonanthine in sorghum induced by FAW larvae are involved in the defense of host plants against insects. However, the compounds involved in host defense and their function need to be further investigated. The functional identification of sorghum secondary compounds against FAW would be conducive to understanding the molecular basis of the coevolution between polyphagous insects and host plants.

## 5. Conclusions

In summary, this study investigated the chemical defense mechanism of sorghum against *S. frugiperda*. Feeding sorghum reduced the fitness of FAWs. The sorghum ingested by the FAWs observably changed the response spectra of secondary metabolites. Gambogenic acid and chimonanthine were specifically present and induced in sorghum, and have a negative impact on FAW larval feeding behavior and larval growth. The current findings have significant implications for the identification of potential natural insecticidal compounds and provide a theoretical basis for the development of environmentally friendly strategies for pest control.

## Figures and Tables

**Figure 1 insects-16-00218-f001:**
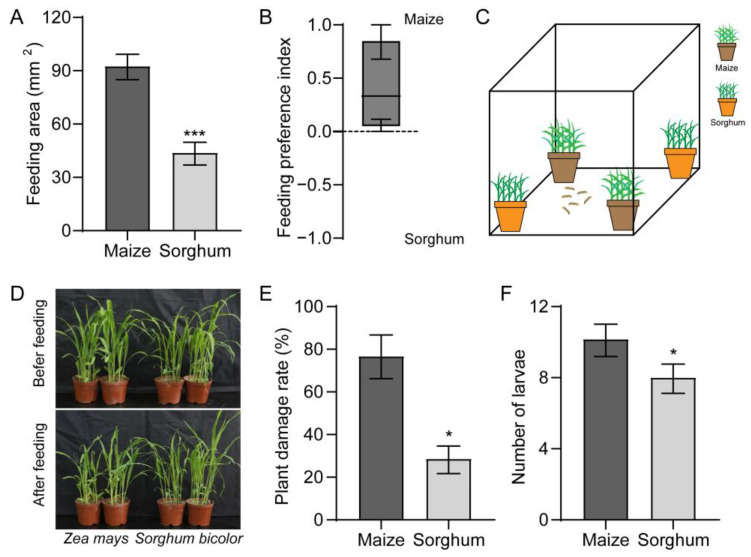
Preferences of *Spodoptera frugiperda* larvae towards maize and sorghum. (**A**,**B**) Feeding area (**A**) and feeding preference index (**B**) of larvae on maize and sorghum leaves. *n* = 24. (**C**,**D**) Schematic diagram (**C**) and representative image (**D**) of the feeding effects of larvae on maize and sorghum plants. (**E**) Damage rate on maize and sorghum plants after being infested by larvae. *n* = 4. (**F**) Number of larvae collected on maize and sorghum. *n* = 3. Data are presented as mean ± SEM. * *p* < 0.05, *** *p* < 0.001.

**Figure 2 insects-16-00218-f002:**
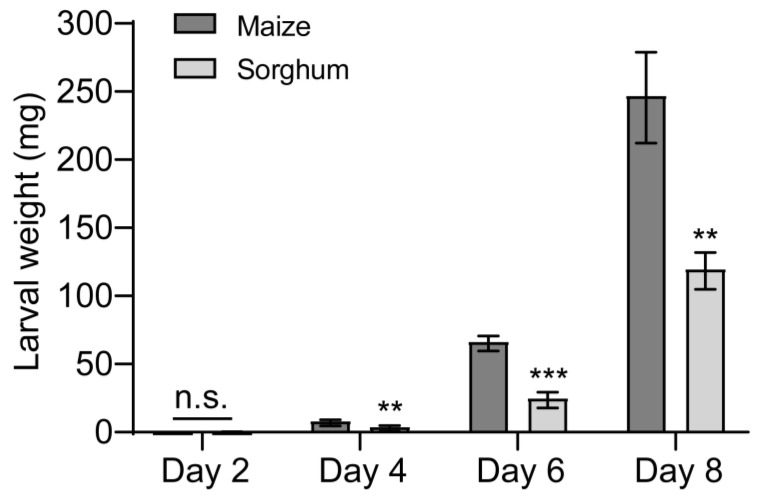
Effects of maize and sorghum on *Spodoptera frugiperda* larval weight. The newly hatched *S. frugiperda* larvae were fed on the leaves of maize and sorghum, respectively, and the larvae were weighed every 2 days. *n* = 5. Five larvae were used for each replicate. Data are presented as mean ± SEM. ** *p* < 0.01, *** *p* < 0.001. n.s. indicates no significant difference (*p* > 0.05).

**Figure 3 insects-16-00218-f003:**
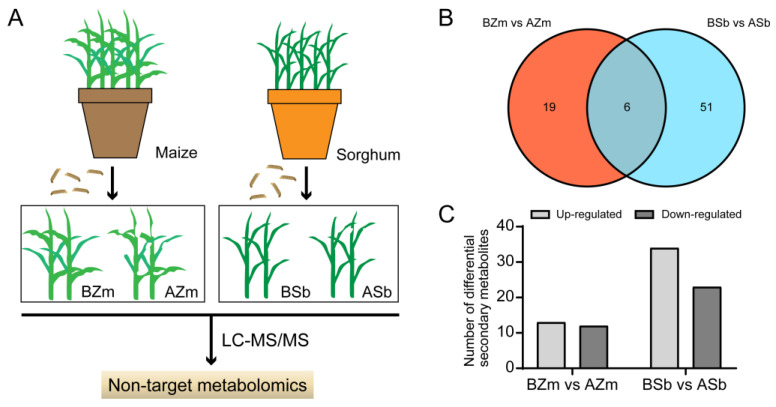
Detection of differential secondary metabolites of maize and sorghum after feeding by *Spodoptera frugiperda* larvae. (**A**) Flow diagram of detection of differential secondary metabolites (DSMs) in maize and sorghum. (**B**) Venn diagrams of DSMs among different maize and sorghum samples. (**C**) Numbers of induced and reduced DSMs in the maize and sorghum groups. BZm, maize plant before feeding by larvae; AZm, maize plant after feeding by larvae; BSb, sorghum plant before feeding by larvae; ASb, sorghum plant after feeding by larvae.

**Figure 4 insects-16-00218-f004:**
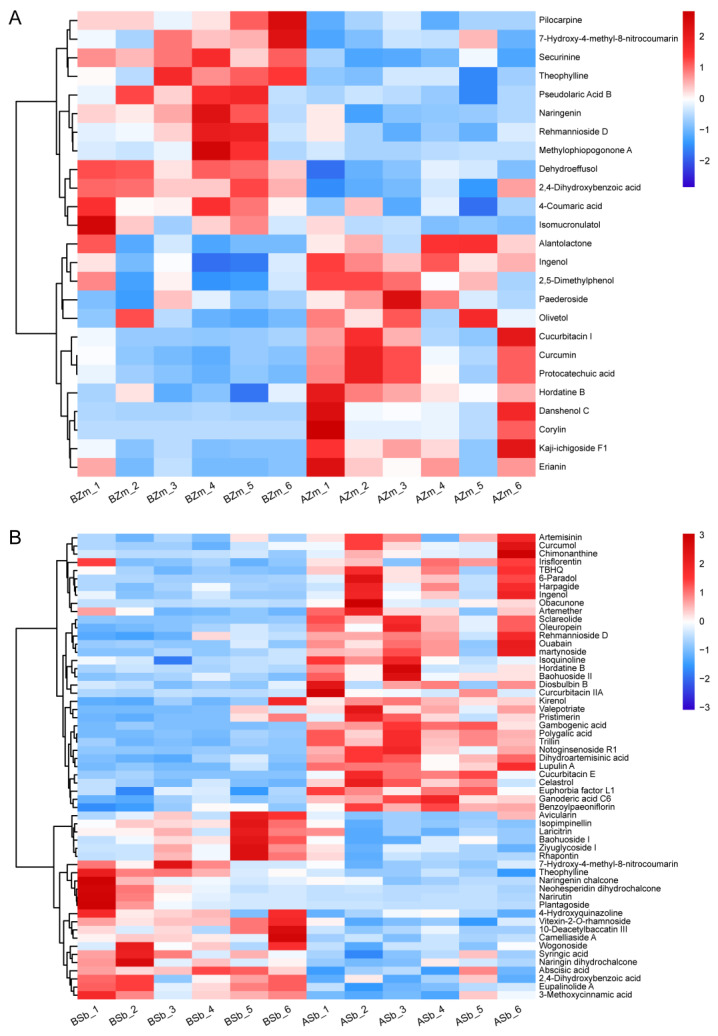
Hierarchical cluster analysis heatmap for the differential secondary metabolites of maize and sorghum. The heatmap of changes in DSMs from maize (**A**) and sorghum (**B**) before and after being fed on by *S. frugiperda* larvae was made by using non-target metabolomics. BZm, maize plant before feeding by larvae; AZm, maize plant after feeding by larvae; BSb, sorghum plant before feeding by larvae; ASb, sorghum plant after feeding by larvae. The horizontal axis represents the samples. The abundance of DSMs in each sample is shown in different colors. Red indicates high abundance, and blue indicates low abundance.

**Figure 5 insects-16-00218-f005:**
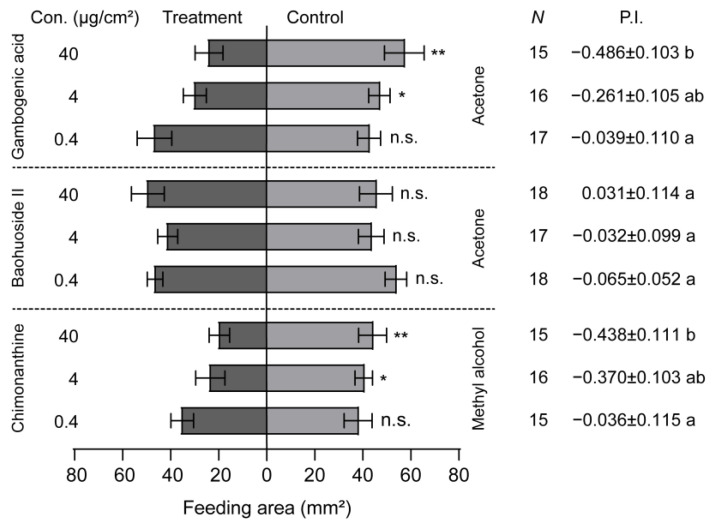
Feeding responses of *Spodoptera frugiperda* larvae to three secondary compounds. The feeding preference of 5th-instar larvae of *S. frugiperda* to gambogenic acid, baohuoside II, and chimonanthine ranged from 0.4 to 40 μg/cm^2^. P.I., preference index. Data are presented as mean ± SEM. * *p* < 0.05, ** *p* < 0.01. Different letters labeled indicate significant differences and n.s. indicates no significant difference (*p* > 0.05).

**Figure 6 insects-16-00218-f006:**
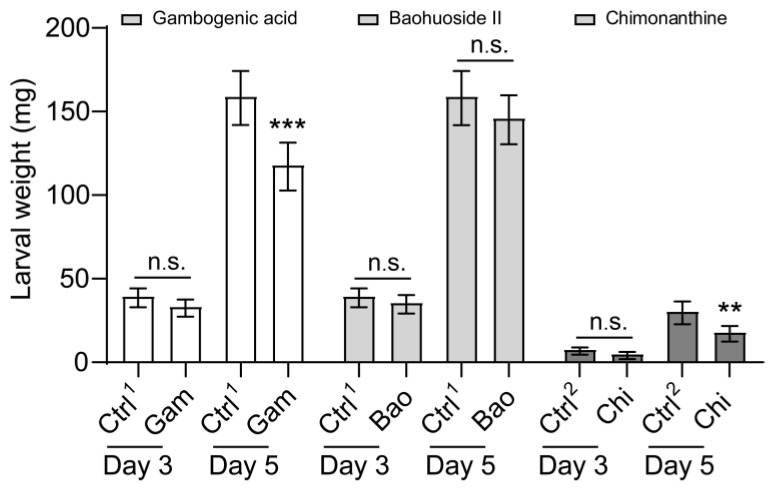
Effect of three secondary compounds on the larval growth rate of *Spodoptera frugiperda*. The growth rate of 2nd-instar *S. frugiperda* larvae to gambogenic acid, baohuoside II, and chimonanthine on day 3 and day 5. Gam, gambogenic acid; Bao, baohuoside II; Chi, chimonanthine. Ctrl^1^, acetone; Ctrl^2^, methyl alcohol. Data are presented as mean ± SEM. The nonparametric test (Mann–Whitney U test) was used. ** *p* < 0.01, *** *p* < 0.001; n.s. indicates no significant difference between control and treatment (*p* > 0.05). The colors white, grey, and black represent gambogenic acid, baohuoside II, and chimonanthine, respectively.

**Table 1 insects-16-00218-t001:** The fold change and VIP value of differential secondary metabolites in maize and sorghum groups.

Class	Name	RT (min)	BZm vs. AZm		BSb vs. ASb	
			FC	VIP	FC	VIP
Terpenes	Diosbulbin B	5.168	–	–	1.45	1.23
	Danshenol C	5.267	12.58	3.04	–	–
	Artemisinin	5.568	–	–	1.54	1.25
	Paederoside	5.571	1.78	1.43	–	–
	Obacunone	6.017	–	–	3.92	2.09
	Eupalinolide A	6.168	–	–	0.40	1.40
	Ouabain	6.293	–	–	2.91	1.16
	Artemether	6.309	–	–	1.70	1.69
	Valepotriate	6.375	–	–	2.83	2.73
	Pseudolaric acid B	6.421	0.58	1.29	–	–
	Dehydroeffusol	6.478	0.41	1.13	–	–
	Curcumol	6.871	–	–	1.72	1.75
	Harpagide	6.986	–	–	3.31	2.28
	Kaji-ichigoside F1	7.558	4.93	1.70	–	–
	Kirenol	7.592	–	–	1.65	1.40
	Sclareolide	7.625	–	–	4.77	3.12
	Ingenol	7.641	1.73	1.04	2.18	1.10
	Cucurbitacin I	7.973	3.41	2.05	–	–
	Polygalic acid	8.356	–	–	2.01	1.96
	Trillin	8.383	–	–	1.96	1.80
	Euphorbia factor L1	8.436	–	–	2.38	1.60
	Curcurbitacin IIA	8.561	–	–	3.18	1.76
	Dihydroartemisinic acid	8.624	–	–	1.84	2.74
	10-Deacetylbaccatin III	8.810	–	–	0.22	1.27
	Ziyuglycoside I	9.082	–	–	0.49	1.31
	Ganoderic acid C6	9.197	–	–	2.88	1.30
	Oleuropein	9.233	–	–	3.12	1.72
	Rehmannioside D	9.304	0.57	1.46	1.73	1.41
	Benzoylpaeoniflorin	9.334	–	–	1.56	1.02
	Cucurbitacin E	9.430	–	–	5.35	2.58
	Notoginsenoside R1	9.684	–	–	12.89	2.74
	Abscisic acid	9.771	–	–	0.72	1.58
	Alantolactone	10.079	1.49	1.39	–	–
	Celastrol	10.769	–	–	2.91	2.17
	Pristimerin	11.149	–	–	2.03	1.45
Simple phenols	7-Hydroxy-4-methyl-8- nitrocoumarin	1.527	0.46	1.46	0.32	2.11
	Isopimpinellin	2.197	–	–	0.34	2.19
	Syringic acid	5.118	–	–	0.57	1.17
	4-Coumaric acid	5.241	0.54	2.00	–	–
	2,4-Dihydroxybenzoic acid	5.386	0.46	2.40	0.51	1.73
	3-Methoxycinnamic acid	5.429	–	–	0.70	1.86
	2,5-Dimethylphenol	5.828	1.34	1.34	–	–
	Rhapontin	6.028	–	–	0.52	1.22
	6-Paradol	6.467	–	–	5.22	1.46
	Erianin	6.483	2.12	1.71	–	–
	Lupulin A	6.854	–	–	4.94	3.04
	Protocatechuic acid	7.359	1.70	2.29	–	–
	TBHQ	7.507	–	–	2.99	1.35
	Curcumin	7.546	1.65	1.63	–	–
	Olivetol	7.556	1.66	1.51	–	–
	Martynoside	8.463	–	–	6.44	2.21
Flavonoids	Corylin	4.783	29.11	2.90	–	–
	Laricitrin	4.959	–	–	0.57	1.45
	Naringin dihydrochalcone	5.211	–	–	0.39	2.36
	Methylophiopogonone A	5.217	0.14	1.55	–	–
	Wogonoside	5.338	–	–	0.48	1.37
	Camelliaside A	5.399	–	–	0.53	2.50
	Naringenin	5.495	0.46	1.62	–	–
	Narirutin	5.717	–	–	0.15	1.16
	Isomucronulatol	5.729	0.60	1.96	–	–
	Irisflorentin	5.794	–	–	2.10	1.06
	Plantagoside	5.842	–	–	0.09	1.07
	Naringenin chalcone	5.872	–	–	0.41	1.89
	Neohesperidin dihydrochalcone	5.955	–	–	0.16	1.09
	Baohuoside I	6.314	–	–	0.66	1.19
	Avicularin	6.337	–	–	0.33	1.45
	Gambogenic acid	8.842	–	–	6.73	2.65
	Baohuoside II	8.982	–	–	3.55	2.27
	Vitexin-2-*O*-rhamnoside	9.484	–	–	0.44	1.52
Alkaloids	Securinine	1.322	0.24	2.46	–	–
	Theophylline	1.347	0.48	1.69	0.36	2.10
	4-Hydroxyquinazoline	3.670	–	–	0.43	1.37
	Pilocarpine	4.972	0.42	1.20	–	–
	Isoquinoline	5.480	–	–	1.62	1.26
	Chimonanthine	6.057	–	–	3.57	2.01
	Hordatine B	8.288	1.97	1.32	4.08	1.88

Note: RT, retention time. BZm, maize plant before feeding by larvae; AZm, maize plant after feeding by larvae. BSb, sorghum plant before feeding by larvae; ASb, sorghum plant after feeding by larvae.

## Data Availability

The data presented in this study are available in the article/Appendix A. Further inquiries can be directed to the corresponding author.

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
