# Peer review of "Fall Armyworm-Induced Secondary Metabolites in Sorghum Defend Against Its Attack"

_insects, 2025, doi:10.3390/insects16020218_

Round 1

Reviewer 1 Report

Comments and Suggestions for Authors

This study investigates the chemical defense mechanisms of sorghum against Spodoptera frugiperda (fall armyworm, FAW), a major agricultural pest in China. The authors found that larvae prefer maize over sorghum. Moreover sorghum reduced larval weight. The study identifies species-specific differential secondary metabolites (DSMs) in both maize and sorghum in response to FAW feeding, with gambogenic acid and chimonanthine found to deter larval feeding and lowering their weight increase. These findings suggest that the present and induced secondary compounds in sorghum have a role in the chemical defense against FAW. This study provides valuable insights into sorghum's chemical defenses and potential bioactive plant compounds for pest management. This study is logic and the results are described clearly, so it’s worth for publication. However, I have several concerns:

1.     In the title "Fall armyworm-induced secondary metabolites in sorghum defend against its attack," maize is not mentioned. Moreover when the authors screened defensive compounds for further testing through two conditions setting (FC > 3 and focusing on flavonoids and alkaloids), the maize group data does not seem to play a significant role in the screening process. So is it necessary to analyze and compare the secondary metabolites before and after feeding on maize in this study?

2.     According to the LC-MS/MS analysis, the number of secondary metabolites in maize is lower than in sorghum. Could this be a contributing factor to the feeding preference? It would be helpful if the authors could discuss this.

3.     In 3.1, the authors studied the effects of feeding different host plants on the growth and development of larvae, but only the weight changes were recorded. It is suggested to also include the development time of the larvae as additional evidence.

4.     In Figures 5 and 6, it’s better that the results for the three secondary metabolites plotted on the same graph. A comparative analysis of the effects of these secondary metabolites should be included in the results, and discuss the possible reasons for the observed differences.

5.     In 3.2 and 3.3, there are several repetitive descriptions about results. The authors are advised to shorten these sections to avoid redundancy.

6.     In the lines 446-455, these sentences primarily describe the results. It is suggested to remove this section to avoid repetition of the results.

7.     In Figure 3B, it is recommended that the authors use more distinct colors to differentiate between the two data groups for better clarity.

Comments on the Quality of English Language

The authors' descriptions of many of the results are repetitive, and the English could be improved to more clearly express the research.

Reviewer 2 Report

Comments and Suggestions for Authors

In this work, the authors analyzed changes in the production of secondary metabolites in sorghum and corn plants in response to FAW attack. They also evaluated FAW preference between sorghum and corn, feeding rates on both hosts and the effect on weight gain in larvae. Furthermore, based on changes in the production of secondary metabolites, they identified compounds that negatively affect the behavior and biological parameters of the pest. The work presents a relevant contribution to future research on new FAW control tactics.

English writing can be greatly improved and it is strongly recommended to have the text reviewed by an English editing service. Several parts of the methodology are unclear and require further explanation. Specific recommendations are presented below:

Line 23: damage rates of sorghum, and suppressing larval weight relative to maize: change "suppressing" by "reducing".  "suppressing" would imply that there is no weight gain at all.

Line 38: How were the compound profiles obtained? A brief methodological description is important in the summary.

Line 40: Briefly describe the biological assay in this section. A brief methodological description should be presented in the summary.

Line 44: change "chemicals" to "metabolites". 

Lines 117-126: In this section, the objectives of the work should be mentioned, but results should not be included.

Line 133: "20% -40 % relative humidity": Are you sure? According to other studies, this humidity is far from the optimal range for the species.

Line 145: "What experimental designs were used in the biological assay experiments? Completely randomized design?

Lines 149-151: How many repetitions were used?

Line 156: "Second, 40 uniform 5th instar larvae as the source were placed in the center of a closed" as the source? It is not clear what this refers to.

Lines 159-160: Do you mean that the larvae selected the host plant? Or were they placed randomly? If the larvae selected the plant, there is no point in saying that it was random, since there are surely mechanisms for detecting volatile compounds that mediate this selection.  This wording is not clear. Review.

Line 160: How long did the experiment last?

Line 161: Do you mean the number of larvae that selected corn or sorghum? The wording is not clear.

Line 165: What does "non-selected conditions" mean?

Lines 175-176: After 24 h, the leaves of maize and sorghum after and before being fed by FAW larvae were quickly cut out..": You mean maize and sorghum leaves that were fed and not fed by larvae? How is it possible to take samples of leaves "before" feeding, if these samples were taken 24 hours after infestation? This methodology is not clear.

Line 216: Díaz Napal's work includes several essays of choice. The authors should clarify which essay they are referring to and what the methodological changes were.

Line 244: "Different letters indicate significant differences.": according to which multiple comparison test?

Figure 1: These photographs do not show any appreciable difference.

Line 263: What does "non selective conditions" mean?

Lines 280-281: "We analyzed the changes of maize secondary compounds before and after feeding by larvae (named AZm vs BZm group)": In the text, "before" comes first, and then "after", so it would be better to put the respective order in the abbreviations: BZm and AZm

Figure 4: Indicate what the X-axis means in this figure. Samples?

Figure 5: What does "P.I." mean? Preference Index? It should be indicated in the figure legend

Figure 6: The letters correspond to groups of different averages according to which multiple comparison test?

Line 480: "sensation" is not an appropriate term in this context. Review wording.

There are additional comments in the attached pdf

Comments on the Quality of English Language

English writing can be greatly improved and it is strongly recommended to have the text reviewed by an English editing service. 

Reviewer 3 Report

Comments and Suggestions for Authors

The study titled ‘Fall armyworm-induced secondary metabolites in sorghum defend against its attack’.  The research investigates the chemical defense mechanisms of sorghum against a significant agricultural pest, Spodoptera frugiperda. The research question is interesting. However, the manuscript can be accepted for publication with few major revisions.

Line 156: 40 5th instar larvae? How did the authors prevent cannibalism?. In all the biological assays the authors ignored the important cannibalistic behavior of FAW, which invalidates the results. Seems like no measures were taken to prevent cannibalism like confining the larvae to a specific leaf, which is usually followed when using FAW.

Line 159: change to “the larvae were allowed to randomly select host plants and fed overnight (about 16h) in natural conditions”

Line 169: larvae was weighed every two days…for how long? Until pupation?

Line 172: change ‘infested about’ to ‘infested by about’. 10-15 3rd instar larvae/seedlings seem be high given that FAW is cannibalistic. How did the authors prevent cannibalism?

Line 177: unfed leaves from the same plant is used as control? This invalidates the results. Controls plants usually need to be free from any mechanical and pest damage.

Figures: please show the errors bars on both sides in all figures.

Line 280, 283: Please assign the abbreviations clearly next to the true synopsis. for instance, before feeding by larvae (BZm) and after feeding by larvae (AZm).

Round 2

Reviewer 2 Report

Comments and Suggestions for Authors

The authors have provided clarifications and made the necessary changes. I consider that the text has improved significantly and is ready to be published.

Author Response

Thank you very much for your comment, which agree with the revised manuscript published in Insects. 

Reviewer 3 Report

Comments and Suggestions for Authors

The authros made significant changes, which helped to improve the manuscript. The manuscript can be accepted for publication in the present form.

Author Response

Thank you so much for your comment. You agree that the revised manuscript can be accepted for publication in Insects.